# Advantages of Induced Circular Dichroism Spectroscopy for Qualitative and Quantitative Analysis of Solution-Phase Cyclodextrin Host–Guest Complexes

**DOI:** 10.3390/ijms25010412

**Published:** 2023-12-28

**Authors:** Márta Kraszni, Balázs Balogh, István Mándity, Péter Horváth

**Affiliations:** 1Department of Pharmaceutical Chemistry, Semmelweis University, Hőgyes Endre utca 9, 1092 Budapest, Hungary; kraszni.marta@semmelweis.hu; 2Department of Organic Chemistry, Semmelweis University, Hőgyes Endre utca 7, 1092 Budapest, Hungary; balogh.balazs@semmelweis.hu (B.B.); mandity.istvan@semmelweis.hu (I.M.)

**Keywords:** circular dichroism spectroscopy, 2D ROESY NMR spectroscopy, quantum chemical calculation, molecular dynamic simulation, nimesulide, fenoprofen, fenbufen, bifonazole, complex stability

## Abstract

The presence of a chiral or chirally perturbed chromophore in the molecule under investigation is a fundamental requirement for the appearance of a circular dichroism (CD) spectrum. For native and for most of the substituted cyclodextrins, this condition is not applicable, because although chiral, cyclodextrins lack a chromophore group and therefore have no characteristic CD spectra over 220 nm. The reason this method can be used is that if the guest molecule has a chromophore group and this is in the right proximity to the cyclodextrin, it becomes chirally perturbed. As a result, the complex will now provide a CD signal, and this phenomenon is called induced circular dichroism (ICD). The appearance of the ICD spectrum is clear evidence of the formation of the complex, and the spectral sign and intensity is a good predictor of the structure of the complex. By varying the concentration of cyclodextrin, the ICD signal changes, resulting in a saturation curve, and from these data, the stability constant can be calculated for a 1:1 complex. This article compares ICD and NMR spectroscopic and molecular modeling results of cyclodextrin complexes of four model compounds: nimesulide, fenbufen, fenoprofen, and bifonazole. The results obtained by the different methods show good agreement, and the structures estimated from the ICD spectra are supported by NMR data and molecular modeling.

## 1. Introduction

The range of applications for cyclodextrins (CyDs) is expanding every day. For food, cosmetic, and especially pharmaceutical products for human use, it is very important to have complete, scientifically based information on the CyD used and its role in the product. This is particularly important for medicinal products for human use. In medicinal products, individual CyDs may be included as orphan drug [1], processing excipients [2], or elements affecting the pharmacokinetic properties of the active substance [3] and also as taste-masking agents. Comprehensive quality assurance and quality control documentation is essential for regulatory GMP (Good Manufacturing Practice) and GLP (Good Laboratory Practice) compliance.

Research/development, in-process control, and quality control at various stages of production require the use of reliable analytical methods. Here, we can think of both solution-phase and solid-phase analytical methods and sometimes heterogeneous systems. A good summary of analytical methods and the synthesis of different semisynthetic derivatives of CyDs is given in the work of Sarabia-Vallejo and coworkers [4].

Infrared [5], Raman [6], and NMR spectroscopy [7], various thermoanalytical techniques [8], and X-ray crystallography for powder X-rays and single crystals can be used to investigate the solid phase. The simultaneous use of several of these may be necessary for a complete characterization [6,9].

Among the solution-phase analytical methods, liquid-state NMR spectroscopy [10,11], which can provide both qualitative and quantitative information, is a prominent example. Qualitative information includes the composition and geometry of the complex, while the stability constant provides a quantitative description of the system. ^1^H (and ^13^C) 1D techniques can be used to determine the composition and stability constant of a drug–CyD complex, while the structure of the host–guest complex can be estimated by 2D ROESY (rotating frame nuclear Overhauser effect spectroscopy) based on the spatial proximity of the host and guest hydrogens. Furthermore, diffusion-ordered spectroscopy (DOSY) offers the possibility to study the polymerization properties. The NMR method has the advantage of small sample requirements (typically 600 μL) and a short measurement time for 1D techniques with sufficiently high-field (>9.4 Tesla, corresponding to >400 MHz ^1^H frequency) magnets. The disadvantage is the relative insensitivity, which can increase the acquisition time for poorly soluble samples, and the significantly increased measurement time for 2D experiments.

Fluorescence spectroscopy [12] is a widely used analytical method because of its great sensitivity and simplicity. Standard fluorescence spectroscopy analyzes the changes of a spectroscopic property, e.g., spectral shift, quantum yield, lifetime, or anisotropy of a fluorescent guest due to complexation.

UV spectroscopy [13] is also often used because it is fast, is cheap, requires relatively small samples, and has usually good sensitivity. Its drawbacks include the need for the presence of a chromophore and the need for a change in the electron distribution of the chromophore group and the energy difference between the HOMO and LUMO orbits due to the interaction with cyclodextrin. The formation of a complex can also cause a bathochromic or hypsochromic shift of a few nm or a change in the intensity of the spectrum. In such cases, the absorbance may increase (hyperchromic effect) or decrease (hypochromic effect) compared to the uncomplexed guest molecule.

Among the separation techniques, HPLC [14] and capillary electrophoresis [15] are the most important, although their main application is in the separation of enantiomers. As a homochiral selector, CyD can be used both as a stationary phase (HPLC, CEC [16]) and as a component of the mobile phase (HPLC, CE). The main shortcoming of separation methods is that they do not directly provide information on the structure of the complex.

Perhaps the least commonly used method to study CyD complexation is circular dichroism (CD) spectroscopy. This may be due to the fact that CD spectroscopy is mostly associated with the structural analysis of proteins; in addition, it is a relatively expensive technique and therefore not widely used. According to a survey, CD spectroscopy is one of the methods considered to be the least important in universities where instrumental analysis is taught [17]. In the professional world, however, it is second to NMR spectroscopy in terms of the information it provides [18].

CD spectroscopy can be considered a relatively simple method—leaving aside the vaguer quantum chemical interpretation-—as it is very similar to UV–vis spectroscopy. The main difference is that the instrument produces alternating right and left circularly polarized light beams at high frequency using a modulator. The molecules with chiral or chirally perturbed chromophore groups absorb the two light components differently, and the difference between the two provides the measured signal [19].
(1)$$∆A=A_{L}-A_{R}=\left(\varepsilon_{L}\times c\times l\right)-\left(\varepsilon_{R}\times c\times l\right)=∆\varepsilon \times c\times l$$
where A_L_ and A_R_ are the absorption of the left and right circularly polarized light components, c is the molar concentration, l is the pathlength, and ε is the corresponding molar absorption.

A major advantage of the technique is that the total absorption of the sample is measured in addition to the CD signal, so that the CD and UV signals of the same material can be detected simultaneously in two channels.

As mentioned above, the presence of the chromophore group is essential for the appearance of the CD spectrum. This condition is not fulfilled for CyDs alone, as they do not have absorption in the analytical range (λ > 220 nm) and therefore do not provide a CD signal, although they are highly chiral molecules. If the guest molecule is not chiral but has a chromophore group, no CD signal is expected. However, complex formation removes this contradiction. If the molecule with the chromophore group and the CyD are in close proximity, the chromophore group becomes chirally perturbed, and the CD signal appears. The phenomenon is called induced circular dichroism (ICD). The formation of a complex is clearly and directly indicated by the ICD signal [20]. Only ROESY can provide this level of direct evidence in a solution-phase assay when the hydrogens of the host and guest molecules are up to about 5 Angstroms apart. The great advantage of the method is that it is easy to calculate the stability constant of the complex, since the measured ICD signal is proportional only to the concentration of the complex. By varying the concentration of the cyclodextrin while keeping the concentration of the guest molecule constant, the intensity of the spectra changes until all the guest molecules are complexed. In this case, a saturation curve can be obtained by depicting the intensity values at a given wavelength as a function of CyD concentration. 

For substituted CyDs, the ICD spectra may differ significantly if the structure of the complex is different as a result of substitution [21]. Furthermore, the ICD spectra of the formed complexes are sensitive not only to the diversity of substituents but also to the degree of substitution and homogeneity [22]. Not only do the spectra provide information about the stability constant but their sign and shape allow the estimation of the complex’ geometry. Based on theoretical considerations, Kodaka [23], Harata [24], and Kajtár [25] found a correlation between the sign of the spectrum and the geometry of the complex. When the molecule is excited by light, the electron transition from the HOMO to the LUMO orbital results in a temporary dipole moment. If the vector of the dipole moment coincides (is parallel) with the axis of the CyD cavity and the molecule is inside of this cavity, a band with a positive sign is obtained; if the vector is perpendicular, the sign becomes negative. If the molecule is located outside of the cavity, exactly the opposite is true. The basis of the principle, known as the Kodaka–Harata rules, is illustrated in Figure 1.

Molecular dynamic (MD) simulations are frequently used in the analysis of host–guest complex formation between CyDs and small molecules. MD methods are often titled as computational (or in silico) molecular “microscopes” because they provide insight into structural and energetic properties of biological systems in water. The process of the host–guest formation and the stability of the complexes can also be investigated through monitoring the trajectories of the simulations. Various software programs (e.g., Desmond), force fields (including OPLS), simulation times, and solvent treatment methods (e.g., the explicit SPC water model used by us) were used in the literature to investigate drug–CyD complexation. A recent review summarizes different aspects of such studies [26].

While MD simulations could be used to predict the structure and stability of CyD complexes, quantum mechanical (QM) calculations are necessary for the estimation of the thermodynamic properties and charges with greater accuracy and for the prediction of other properties, including HOMO–LUMO energies or UV–vis, IR, and NMR spectra. The current state of QM studies on CyD complexes were reviewed by Mazurek and Szeleszczuk in 2022 [27]. According to their survey, while semiempirical methods (e.g., AM1, PM3, PM6, and PM7) and Møller–Plesset perturbation theory (MP2) are sometimes also utilized, the density function theory (DFT) is the most frequently used QM method for the assessment of CyD host–guest complexes. DFT is mostly combined with the B3LYP functional and with a 6-31G* (or higher) basis set. Regarding the environment of the system, the polarizable continuum model (PCM) is the most popular implicit solvent model. 

The aim of the present article is to demonstrate the applicability of CD spectroscopy in the structure determination of drug–CyD complexes. Therefore, complex structures were first estimated by using CD measurements, and then the structures were confirmed by NMR and also by molecular dynamics and quantum chemical calculations. 

## 2. Results

The model compounds chosen for this study are poorly water-soluble drugs whose solubility and bioavailability can be improved by the application of CyDs. Their structures are shown in Figure 2. Each molecule contains at least two phenyl groups, but their connection differs pairwise. Therefore, the discussion of the structures should be divided into two parts. While the common structural elements of fenbufen and bifonazole are the two directly connected phenyl groups (biphenyl structure), fenoprofen and nimesulide both have two phenyl groups connected through an ether oxygen (phenoxyphenyl structure). This structural difference fundamentally affects the geometry of the molecules and consequently the conformation of the CyD complex.

### 2.1. CD and UV Spectroscopy Measurements

#### 2.1.1. Comparison of Bifonazole and Fenbufen

Figure 3 shows the CD (top) and UV (bottom) spectra of bifonazole (3a) and fenbufen (3b). For the CD measurement of compounds containing the biphenyl group, cells with a smaller light path (0.2 cm; 0.1 cm) had to be used because the biphenyl group has a high molar absorption coefficient. The absorption maximum of bifonazole is around 255 nm, and there is no significant wavelength shift due to complexation with β-CyD (BCyD), but it is the only molecule where cyclodextrin complexation causes a decrease in spectral intensity at a constant bifonazole concentration.

The UV spectrum of uncomplexed fenbufen has a maximum at 284.5 nm which is significantly higher than the maximum of bifonazole. This is due to the direct attachment of a carbonyl group to the biphenyl ring. The nonbonding electron pair of the oxo group can conjugate with the aromatic system. There is no significant wavelength shift in the spectrum of fenbufen, but at a constant guest concentration, the intensity of the spectrum increases due to complexation. 

#### 2.1.2. Comparison of Fenoprofen and Nimesulide

Figure 4 shows the CD and UV spectra of fenoprofen (4a) and nimesulide (4b). For the two guest molecules containing the phenoxyphenyl groups, unlike the molecules containing the biphenyl chromophore, a bathochromic and hyperchromic shift is observed in the UV spectra due to complexation with BCyD. Since only the excitation band specific to the guest molecule is selectively present in the complex, no such shift is observed in the CD spectrum. For fenoprofen, the absorption maximum of the uncomplexed molecule is at 270.6 nm, whereas for a 20-fold excess of BCyD, a bathochromic shift of nearly 3 nm is observed in the spectrum. The CD spectra show ellipticity maxima at 274.6 and 280.6 nm, respectively. The UV spectrum of nimesulide has a maximum at 391.4 nm, which shows a bathochromic shift of about 10 nm and a significant increase in intensity due to complexation. This bathochromic shift was already visible during solution preparation, as the solutions became more yellowish with increasing BCyD concentration. This color change indicates that the nitrophenyl-methanesulfonamide group of the molecule is probably involved in the complex formation. At a lower wavelength, around 271 nm, a low-intensity band appears in the CD spectrum, which can be assigned to the phenoxy group. It appears as a single shoulder in the UV spectrum and shows no shift on the wavelength scale with complexation. The weaker intensity band at 271 nm indicates that the position of the phenoxy ring may be close to the CyD edge and rather outside the cavity, and the aromatic ring plane is perpendicular to the CyD axis.

### 2.2. NMR Measurements

Figure 5 shows the 2D ROESY NMR spectrum of the fenbufen–BCyD complex. The most intense cross peaks can be seen between the hydrogens of the aromatic ring closer to the side chain (H1,1′ and H2,2′) and the H3, H5 hydrogens inside the cavity of BCyD and also the H6 of BCyD located at the narrower rim of the CyD. While the H3,3′ hydrogens of the terminal phenyl group still have strong through-space interaction with the H3 hydrogen of BCyD, weak or no cross peaks appear in the cases of H4,4′ and H5 of fenbufen. It should be noted that the ^1^H signals of the latter hydrogens are slightly broadened in the 1D spectrum, which may indicate the flexibility of the terminal phenyl group in the complex.

Figure 6 shows the 2D ROESY NMR spectrum of the nimesulide–BCyD complex. The hydrogens of the trisubstituted phenyl ring show the closest proximity to the BCyD hydrogens inside the cavity. In addition, the H5,5′ hydrogens of the phenoxy ring have intense cross peaks with H3 and H6 of BCyD, while the other hydrogens (H6,6′ and H7) show little or no interactions. The methyl hydrogens of the methanesulfonamide group also have a weak cross peak with H3 of BCyD.

### 2.3. Molecular Dynamic and Quantum Chemical Calculations

Our objective is to calculate transient dipole vectors of the complex excited state and verify the sign of the ICD spectra based on these results.

In our MD simulation, the structures of the complexes spontaneously formed were used as a starting point for modeling. As a result of the MD runs, stable complexes were formed, and the drug compounds spontaneously entered into the BCyD cavity and stayed inside until the end of the simulations. The only exception was the fenbufen–BCyD system: this complex dissociated by the end of the run; therefore, a longer 1000 ns simulation was also completed. By the end of this calculation, a stable complex was formed. In the case of bifonazole, the complex with the biphenyl group inside the ring was formed. The second assembly of the bifonazole (which started with the phenyl group inside the ring) also stayed together during the whole course of the simulation. 

Although the formation of the complexes is mainly determined by lipophilic interactions, we also observed hydrogen-bond interactions between the CyD and the drug molecules. In the case of bifonazole, an alternative coupling was assumed based on the NMR structure [28]. Such a complex was not formed spontaneously (it was preassembled and investigated as an additional MD run) but remained stable during the simulations confirming its existence. The calculated numerical results for the transition dipole moment vectors are given in Table 1.

## 3. Discussion

### 3.1. Bifonazole and Fenbufen

#### 3.1.1. Bifonazole

The benzenoid chromophores of the phenyl and biphenyl groups of bifonazole overlap and cannot be separated in the spectrum. The imidazole, phenyl, and biphenyl groups are arranged in a distorted tetrahedral arrangement around the central sp^3^ hybrid chiral carbon atom. The positive ICD signal suggests two possible arrangements. If the biphenyl group is immersed in the CyD cavity, the biphenyl group will be almost parallel to the axis of the CyD, while the 1-4 carbon atom axis of the phenyl group will be perpendicular. The phenyl group will then be located on the surface of the cavity. The other possibility is that the phenyl group is inside the cavity. In this case, the axis of the biphenyl rings will be perpendicular to the axis of the CyD and will be outside the cavity (Figure 7). This latter hypothesis is supported by the article by Kelemen et al. [28]. Their NMR measurements show that the hydrogens of the phenyl group are spatially close to the H3 and H5 hydrogens of the BCyD. However, our MD simulations support the possibility of both arrangements. 

#### 3.1.2. Fenbufen

Due to the biphenyl group of fenbufen and the conjugation of the γ-oxo group, the absorption maximum is about 30 nm higher than for bifonazole. In the absence of a BCyD, the nonbonding electron pair of the carbonyl oxygen may form a hydrogen bond with the water molecules of the solvent, making it more difficult to excite. The possibility of conjugation increases in an apolar medium, thus increasing the intensity of the UV spectrum in the complexes and hence the intensity of the ICD signal. The positive ICD signal in the absorption-band region of the biphenyl group suggests that the biphenyl group is parallel to the axis of the CyD in the cavity (Figure 8). The cross peaks in the ROESY spectrum also support this hypothesis. Since the H4,4′ and H5 hydrogens of fenoprofen do not show cross peaks with the H3, H5, and H6 of the BCyD, it is likely that they are slightly ‘out of the cavity’. In addition, the more intense cross peaks between the hydrogens of the phenyl ring closer to the side chain and the H5 and H6 hydrogens of the BCyD and the less intense cross peaks between the hydrogens of the terminal aromatic ring and the same BCyD hydrogens, and at the same time their more intense interaction with H3, suggest that the molecule is immersed in the cavity, with the terminal ring towards the wider rim of the BCyD. In the case of fenbufen, the stability constant of the BCyD complex was previously determined by NMR spectroscopy [29], but no 2D ROESY measurements were available to estimate the structure.

### 3.2. Fenoprofen and Nimesulide

In the case of UV spectroscopy, the spectra of fenoprofen and nimesulide are showing batho- and hyperchromic shifts due to complexation. The fact that the maximum values show an increasing shift with increasing concentration of CyD indicates that the UV spectrum is derived from the mole fraction weighted average of the molar absorption coefficient of the uncomplexed and complexed guest molecules. In contrast, there is no spectral shift in the CD spectra, where only the absorption band specific to the complexed guest molecule appears.

#### 3.2.1. Fenoprofen

The ICD spectrum of fenoprofen shows a typical structure at the characteristic range of benzenoid bands. In aqueous solution, the nonbonding electron pairs of the ether-bonded oxygen are less able to conjugate with the aromatic system because they form a hydrogen bond with water molecules. Once inside the CyD cavity, the probability of hydrogen-bond formation decreases, and thus, the probability of conjugation increases. Based on the positive sign of the relatively intense CD spectrum, it is certain that the aromatic part of the molecule is predominantly located in the CyD cavity. The logarithm of the stability constant of the complex is close to 3, based on both CD (3.06) and NMR (2.98) measurements [22]. This value represents a stable complex. Since the ICD spectrum is quite structured, it is likely that the conformational stability in the BCyD cavity is quite high. It is also expected that the carboxylate group of fenoprofen will form a hydrogen bond with the hydroxyl groups on carbon atoms 2 and 3 of BCyD, which stabilizes the complex. The ROESY spectrum of the fenoprofen–BCyD complex—published in our earlier paper [22]—shows the most intense cross peaks between the H1 and H5,5′ hydrogens of fenoprofen and the BCyD hydrogens, but it seems that they have similar intensity to all three BCyD hydrogens inside the cavity and near the rim. Based on the intensity of the signals, the H7, H2, and H3 hydrogens are less probably located inside the cavity. Although the earlier published spectrum does not show the methyl hydrogens of fenoprofen, they also have through-space interaction mainly with the H3 hydrogen of BCyD (Appendix A). All of these observations suggest that the structure of the complex may be diverse. The methyl group is located near the wider rim and the terminal part of the phenoxy group near the narrower rim and out of the cavity of BCyD. The complex structure obtained by molecular modeling (Figure 9) also shows that the carbon atom of the phenoxy ring, labelled 7, extends through the cavity, supporting the conclusions drawn from the NMR measurements. 

#### 3.2.2. Nimesulide

In the case of nimesulide, the CD spectrum shows that the nitrophenyl-methanesulfonamide group is dominantly involved in the formation of the ICD spectrum. The increasing yellow color during complexation and the bathochromic shift in the UV spectrum indicate that the methanesulfonamide group is partially deprotonated. As there is no basic functional group in the system, only the formation of a hydrogen bond with the OH groups of BCyD can be considered. In the ROESY spectrum of the nimesulide–BCyD complex (Figure 6), the intense cross peaks of the H3 and H4 signals with the H5 of BCyD show that these hydrogens of the trisubstituted phenyl ring are located inside the cavity of BCyD near the narrower rim, while the H2 and H6,6′ hydrogens are situated closer to the wider rim. The terminal hydrogens of the phenoxy ring (H7 and H6,6′) are mainly out of the cavity. In addition, there is a small possibility that the methyl hydrogens of the methanesulfonamide group and the H3 hydrogen of BCyD may be in close proximity.

The least consistent data were obtained for nimesulide (Figure 10). NMR measurements show that hydrogen 6 of the phenoxy ring gives a weak cross peak with hydrogens 3 and 6 of the BCyD, and hydrogen 7 shows none. However, the molecular dynamics calculations suggest that the substituted nitro-phenoxy-methanesulfonamide group is not very well immersed in the cavity. The reason for this discrepancy may be that the nitro group makes the quantum mechanical calculations less straightforward.

### 3.3. Transition Dipole Moment Vectors

Overall, it can be stated that the quantum mechanics and molecular dynamics calculations and also NMR measurements support our hypothesis of the complex geometries based on CD measurements. The transition dipole moment vectors obtained from the calculations and the sign of the CD spectra are consistent with the Kodaka–Harata rules. In the case of bifonazole, when the phenyl ring is immersed in the cavity, the biphenyl group is outside the cavity and its excitation dipole moment is almost perpendicular to the BCyD axis. In the other cases, the excitation dipole moment vectors are locked to the axis of the BCyD by 15–30 degrees, so that the appearance of a CD band with a positive sign is also justified in these cases (Figure 11).

## 4. Materials and Methods

### 4.1. Materials

β-cyclodextrin (BCyD) was the product of Cyclolab Ltd., Budapest, Hungary. Fenbufen (γ-oxo-(1,1′-biphenyl)-4-butanoic acid), fenoprofen calcium salt (calcium (±)-2-(3-phenoxyphenyl)propanoate dihydrate, and nimesulide (N-(4-nitro-2-phenoxyphenyl)methanesulfonamide) were purchased from Sigma-Aldrich/Merck Group, Budapest, Hungary. Bifonazole ((*RS*))-1-[phenyl(4-phenylphenyl) methyl]-1H-imidazole) was the product of Sandoz, Budapest, Hungary. Deuterium oxide (D_2_O, 99.96% D) and sodium-deuteroxide (NaOD 40% w/w solution in D_2_O, 99.5% D) were the products of VWR Chemicals, Leuven, Belgium and Alfa Aesar Chemicals, Kandel, Germany, respectively.

### 4.2. NMR Measurements

One-dimensional ^1^H and 2D ROESY NMR spectra of the fenbufen–BCyD (4 mM fenbufen, 8 mM BCyD in D_2_O dissolved with small amount of NaOD) and nimesulide–BCyD (2 mM nimesulide, 6 mM BCyD in D_2_O dissolved with small amount of NaOD) complexes were recorded at 25 °C on a Varian DDR spectrometer (599.9 MHz for ^1^H, Varian, Palo Alto, CA, USA) equipped with a dual 5 mm inverse detection gradient (IDPFG) probe head with z-gradient. All spectra were processed using MestReNova 14.2, Mestrelab Research, Santiago de Compostela, Spain. ROESY spectra were measured using the ROESYAD pulse sequence with a mixing time of 300 ms and a spectral width of 10 ppm in both dimensions. The spectrum was acquired into 901 complex points in t2 with 32 scans coadded at each of 256 t1 increments. The spectrum was processed to 1024 × 2048 data points. 

NMR measurements of fenoprofen were provided in a previous, closely related article [22].

### 4.3. CD Measurements

CD and UV experiments were performed on a Jasco J-815 spectrometer, Jasco LTD, Tokyo, Japan in cylindrical Hellma cuvettes with pathlength of 0.1 cm for fenbufen and 0.2 cm for bifonazole. For fenoprofen and nimesulide, 1.0 cm cells were used. The slit was set to 2 nm, the registration speed to 50 nm/min, and the accumulation to 5. In order to obtain accurate readings of the measured ICD values, noise filtering was performed using the fast Fourier transform menu in the Spectra Manager program (Jasco Spectra Manager V2.14.06).

#### Preparation of Solutions for CD Measurements

Bifonazole was dissolved in DMSO to form 0.1 M stock solution, which was further diluted to 1.6 × 10^−3^ M for the experiments. The BCyD stock solution was 0.016 M in water. For the solutions to be measured, the bifonazole–BCyD concentration ratios were set to 1:0; 1:5, and 1:25. For fenbufen and nimesulide, the same solutions were prepared as for the NMR measurements but at different drug–BCyD concentration ratios (1:0; 1:5; 1:25; 1:30). Since fenoprofen calcium salt cannot be dissolved in pure water, methanol-*d*_4_ was used to prepare a 4 mM stock solution. BCyD stock solutions were prepared in H_2_O (12 mM). In the solutions used for the measurements, the concentration of fenoprofen was 0.4 mM and the fenoprofen–BCyD concentration ratios were 1:0, 1:5, and 1:20.

### 4.4. Computational Methods

#### 4.4.1. Preparation of the Compounds

The structures of the four drug molecules were imported into the Schrödinger’s Maestro GUI (v.9.0) (Schrödinger Release 2022-2: Maestro, Schrödinger, LLC, New York, NY, USA, 2022) as SMILES codes, hydrogen atoms were added, then quick optimizations were completed with Maestro 3D Builder tools. In the cases of bifonazole and fenoprofen, the R enantiomers were used, as were the cationic form of bifonazole (protonated imidazole ring nitrogen), the anionic forms of fenbufen and fenoprofen (deprotonated carboxylic groups), and both the neutral and anionic forms of nimesulide (deprotonated sulfonamide group). A conformational search was completed for each drug compound with molecular mechanics (MM) using the Schrödinger’s MacroModel module (Schrödinger Release 2022-2: MacroModel, Schrödinger, LLC, New York, NY, USA, 2022) using an OPLS4 forcefield using the default settings with water solvation (analytical generalized-born/surface-area, GB/SA) model. The lowest energy conformers of the compounds were used for further modeling.

#### 4.4.2. Preparation of β-Cyclodextrin

The GETPAW X-ray crystallography structure was downloaded from the Cambridge Structural Database (https://www.ccdc.cam.ac.uk/ (accessed on 7 November 2023)) [30,31]. This structure contained a dimer of a BCyD (R)-(-)-fenoprofen clathrate hydrate complex (with the fenoprofen molecule inside the BCyD ring) [32,33]. Only one half of the dimer was used, all water molecules were removed, some missing hydrogens were added, and a restrained minimization was applied (only for the hydrogen atoms), and finally, the fenoprofen molecule was also removed. The remaining single BCyD structure was used as input for the molecule dynamics (MD) simulations.

#### 4.4.3. Molecule Dynamic Simulations 

Desmond software (https://newsite.schrodinger.com/platform/products/desmond/ accessed on: 20 December 2023), Schrödinger Release 2022-2: Desmond Molecular Dynamics System, D. E. Shaw Research, New York, NY, USA, 2022; Maestro-Desmond Interoperability Tools, Schrödinger, New York, NY, USA, 2022 was used for the molecular dynamics (MD) simulations. Desmond’s System Builder application was used to assemble the setup for the runs: a 30 × 30 × 30 Å water box (explicit SPC water model) was built, always with a single BCyD and a single drug molecule (1:1 ratio). The BCyD ring was positioned roughly in the middle of the box, and the drug molecules were positioned about 5 Å distance from the center of the BCyD ring; the plane of the ring was perpendicular to the drug–BCyD axis. In the case of the bifonazole molecule, another setup was assembled in which the single phenyl ring was positioned inside the BCyD ring. Prior to the dynamics, a short (0.10 ns) Desmond minimization was completed with the default settings. Desmond dynamics with 100 ns simulation time were completed with an NPT ensemble at 27 °C (300 K) temperature with normal 101,325 Pa (1.01325 bar) pressure; the recording interval for the trajectory was 100 ps. Model systems were relaxed at the beginning of the runs according to Desmond’s standard procedure. A longer 1000 ns run was also completed with the fenbufen–BCyD system. A second 100 ns simulation was performed with the bifonazole molecule with the above-mentioned second assembly.

#### 4.4.4. Quantum Mechanics Calculations

Jaguar software (v.11.6) was used for the quantum mechanics (QM) calculations (Schrödinger Release 2022-2: Jaguar, Schrödinger, LLC, New York, NY, USA, 2022). The lowest energy conformers of the MM searches and the last steps of the trajectories were used as input for the QM calculations. All calculations were completed using the density function theory (DFT) B3LYP-D3 method with 6-31G** basis set in implicit PCM water model. In the case of the MM conformers, full geometry optimizations were completed along with single-point-energy (SPE) calculations for the BCyD–drug complexes. These were followed by other SPE calculations completed with the optimization of the first excited states with time-dependent density functional theory (TDDFT) with the Tamm–Dancoff approximation [34].

## 5. Conclusions

CD spectroscopy can be used to study BCyD complexes when a suitable chromophore group is present. The ICD signal is **clear evidence** of the complex formation in the case of achiral molecules or racemic mixtures. This method has inherent selectivity that other spectroscopic methods lack, as only the complex provides a signal in the ICD spectrum. The ICD spectra allow the estimation of the complex structure. When combined with MD and QM calculations, the results are almost as informative as those obtained by NMR spectroscopy. As previously described [21,22], CD spectroscopy can be used to calculate the stability constant of the complex, and ICD spectra can highlight changes in complex geometry when using different substituted CyDs.

Obviously, like all methods, CD spectroscopy has its drawbacks. If the ∆ε is small compared to the high molar absorption (ε) of the chromophore, the signal-to-noise ratio may be poor. If the stability constant of the complex is low (log*K* < 2) or the geometry of the complex is unfavorable, no or only a weak and noisy ICD signal can be detected.

## Figures and Tables

**Figure 1 ijms-25-00412-f001:**
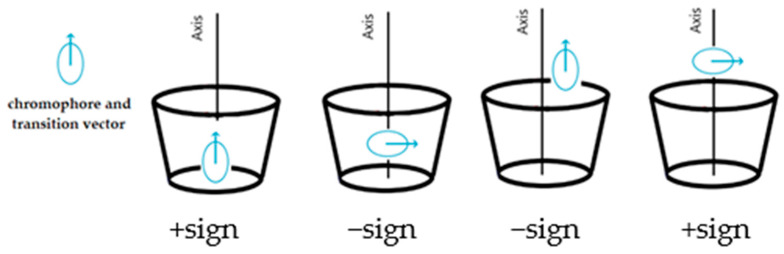
A brief summary of the Kodaka–Harata rules.

**Figure 2 ijms-25-00412-f002:**
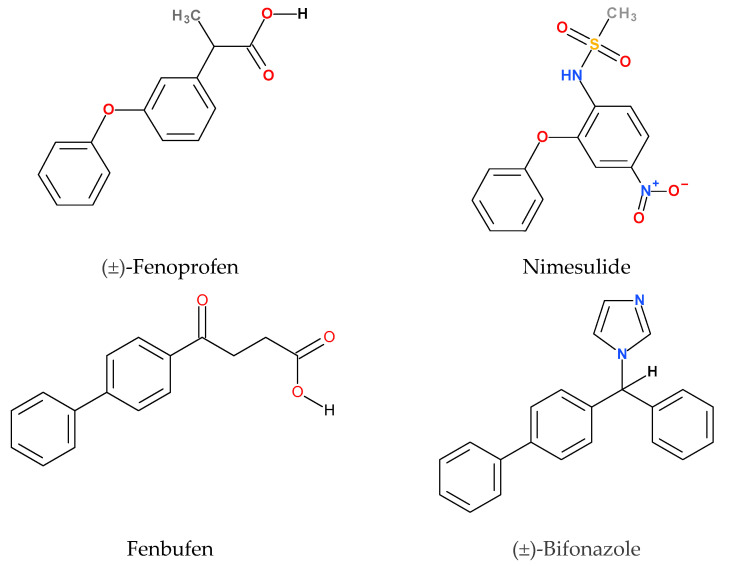
Structures and common names of model compounds. The official IUPAC names are given in Section 4.1.

**Figure 3 ijms-25-00412-f003:**
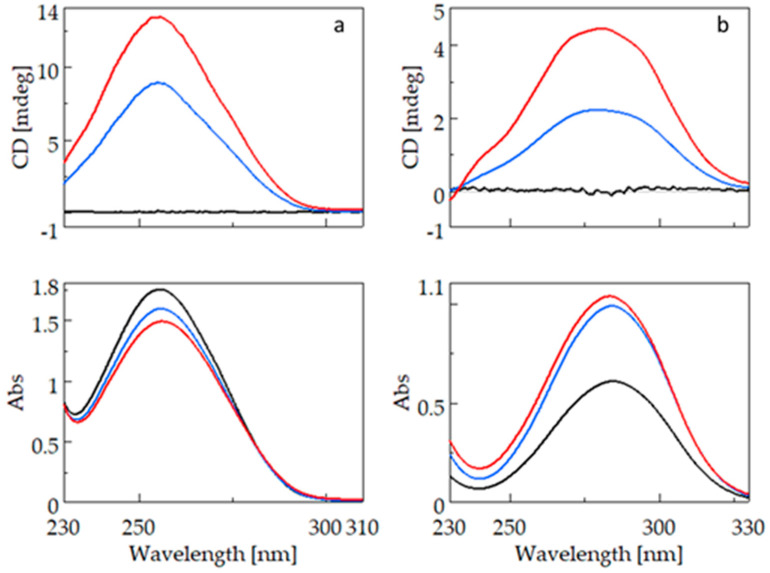
CD and UV spectra of bifonazole (**left**) and fenbufen (**right**) and their β-cyclodextrin complexes. (**a**): CD and UV spectra of bifonazole (black), bifonazole–β-cyclodextrin at a concentration ratio of 1:5 (blue), and bifonazole–β-cyclodextrin at a concentration ratio of 1:25 (red). (**b**): CD and UV spectra of fenbufen (black), fenbufen–β-cyclodextrin at a concentration ratio of 1:5 (blue), and fenbufen–β-cyclodextrin at a concentration ratio of 1:25 (red).

**Figure 4 ijms-25-00412-f004:**
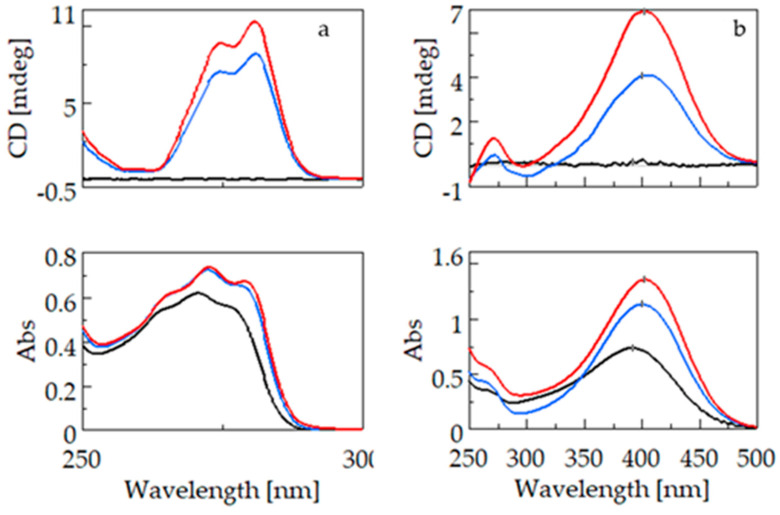
CD and UV spectra of fenoprofen (**left**) and nimesulide (**right**) and their β-cyclodextrin complexes. (**a**): CD and UV spectra of fenoprofen (black), fenoprofen–β-cyclodextrin at a concentration ratio of 1:5 (blue), and fenoprofen–β-cyclodextrin at a concentration ratio of 1:20 (red). (**b**): CD and UV spectra of nimesulide (black), nimesulide–β-cyclodextrin at a concentration ratio of 1:5 (blue), and nimesulide–β-cyclodextrin at a concentration ratio of 1:30 (red).

**Figure 5 ijms-25-00412-f005:**
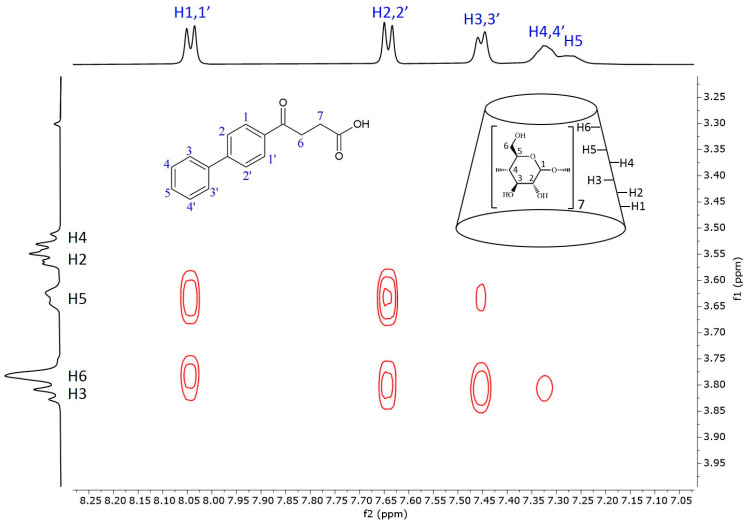
Two-dimensional ROESY NMR spectrum of the fenbufen–BCyD complex in D_2_O. Assignments for fenbufen hydrogens are in blue and for BCyD are in black.

**Figure 6 ijms-25-00412-f006:**
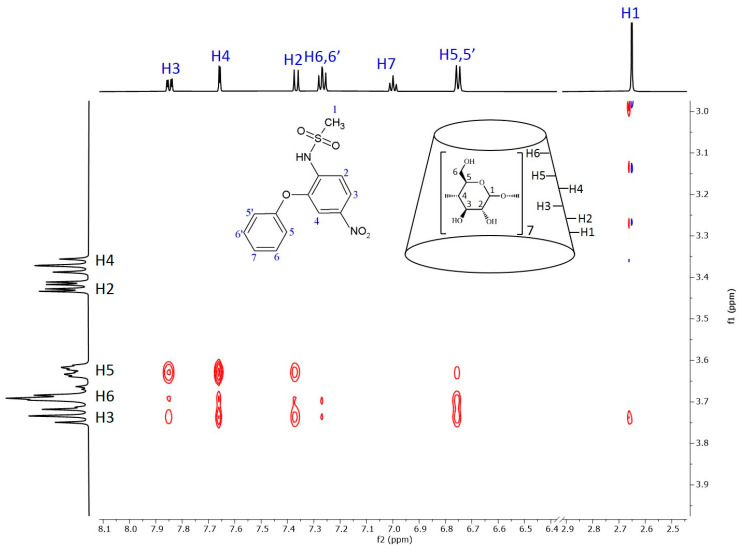
Two-dimensional ROESY NMR spectrum of the nimesulide–BCyD complex in D_2_O. Assignments for nimesulide hydrogens are in blue and for BCyD are in black.

**Figure 7 ijms-25-00412-f007:**
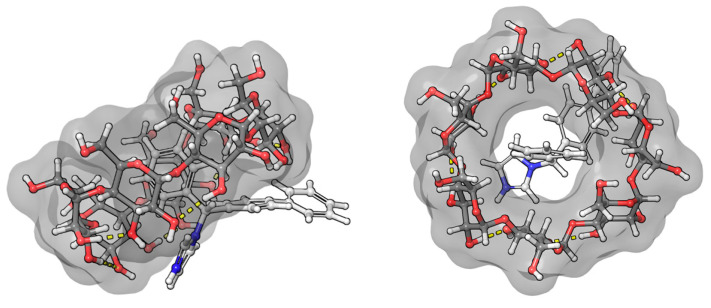
Model of the bifonazole–BCyD complex. See the detailed 3D structure in the Appendix A.

**Figure 8 ijms-25-00412-f008:**
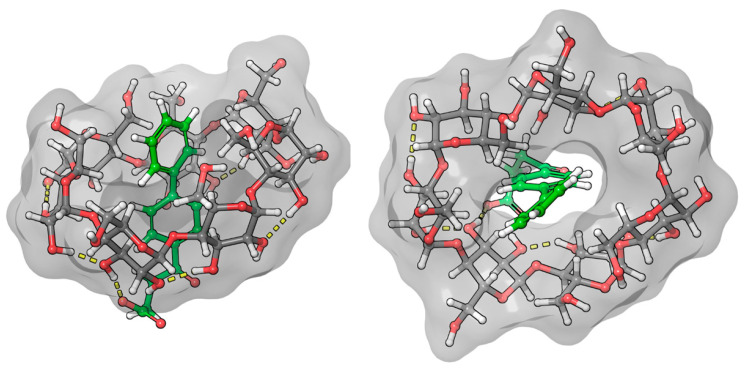
Model of the fenbufen–BCyD complex. See the detailed 3D structure in the Appendix A.

**Figure 9 ijms-25-00412-f009:**
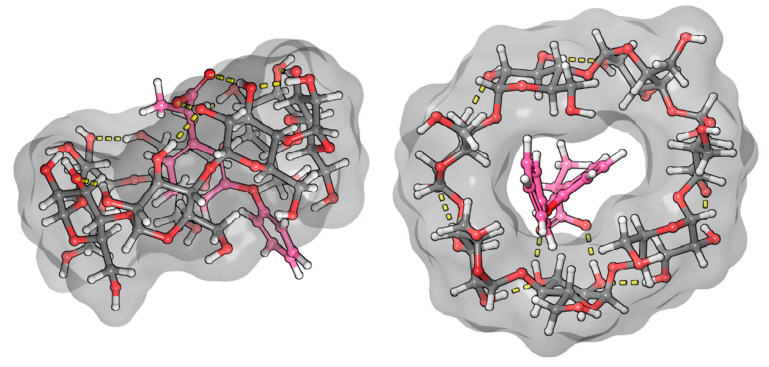
Model of the fenoprofen–BCyD complex. See the detailed 3D structure in the Appendix A.

**Figure 10 ijms-25-00412-f010:**
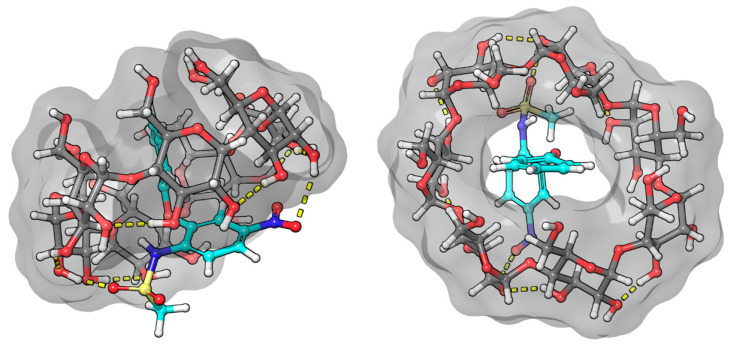
Model of the nimesulide–BCyD complex. See the detailed 3D structure in the Appendix A.

**Figure 11 ijms-25-00412-f011:**
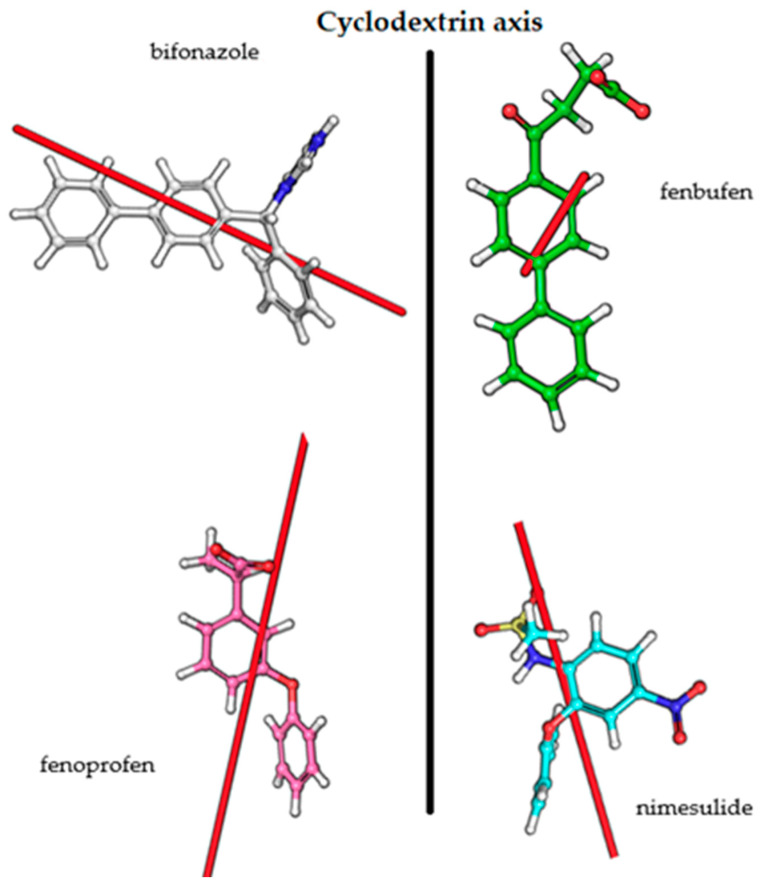
Transition dipole moment vectors for the studied molecules.

**Table 1 ijms-25-00412-t001:** Calculated numerical values of the transition dipole moments.

	Transition Dipole Moment (D)			
	X	Y	Z	Total
Fenoprofen (water)	0.0917	−0.5624	−0.4419	0.7211
Fenoprofen–BCyD	−0.0456	−0.0700	0.1518	0.1732
Fenbufen (water)	−0.9793	−0.8206	−0.4739	1.3627
Febufen–BCyD	0.0105	−0.1213	0.2067	0.2398
Nimesulide (water)	0.5110	0.2802	0.2288	0.6261
Nimesulide–BCyD	−0.5166	−0.3848	−1.3888	1.5309
Bifonazole (water)	−0.3578	−0.2423	−0.0594	0.4361
Bifonazole–BCyD biphenyl in ring	−0.0514	0.7572	−0.7827	1.0902
Bifonazole–BCyD phenyl in ring	−1.3510	1.9622	−3.2134	4.0002

## Data Availability

Data are contained within the article or Appendix A.

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
