# Peer review of "Advantages of Induced Circular Dichroism Spectroscopy for Qualitative and Quantitative Analysis of Solution-Phase Cyclodextrin Host–Guest Complexes"

_ijms, 2023, doi:10.3390/ijms25010412_

Round 1

Reviewer 1 Report

Comments and Suggestions for Authors

The manuscript submitted for review is not good, too superficially dealt with the topic. I did not find in the manuscript a justification of why exactly these compounds were dealt with, it is only stated that they are model compounds. The complexes obtained have a stoichiometry of 1:1, while in the text and in the figure caption there are other stoichiometries, they refer to mixtures and not complexes, although the authors call complexes mixtures in the ratio of 1:5. The introduction to the manuscript is too long and previous studies on the subject are not presented. Instead, there is a theoretical introduction as in the textbook. The introduction does not explain the abbreviations (LADME, GMP and GLP). They are not in common use, like NMR, so they should be explained. The title of the manuscript ends with a period. We do not put periods at the end of titles. The unit MHz, a unit of frequency, does not refer to a magnetic field, but to an electromagnetic wave (l. 58 and 59). In l. 67, instead of "Cyd", the abbreviation "CyD" should be used. In the caption of Figure 3, the designation of the type of cyclodextrin is missing (l. 157). In l. 163, the comma after "Figure 4" should be omitted. In the figures in the main body of the manuscript and in the supporting materials, the stereochemistry on the C1-O bond should be indicated; since the geometry of the bonds is indicated at all, this should be done consistently. In the caption of Table 1, instead of the full name of the unit "debye" there should be the abbreviation "D".  In the supporting materials, correct the subscript in the water formula and in the deuterated methanol designation. 

Comments on the Quality of English Language

The language is not the best. The choice of words is not always correct. I suggest that a language correction be done. 

Author Response

The manuscript submitted for review is not good, too superficially dealt with the topic. I did not find in the manuscript a justification of why exactly these compounds were dealt with, it is only stated that they are model compounds.

The manuscript aims to demonstrate the importance of ICD spectroscopy using model compounds. It provides both theoretical and experimental data. The model compounds used are poorly water-soluble drug substances, which can have their solubility and bioavailability improved by cyclodextrin. Various non-steroidal anti-inflammatory drugs (NSAIDs) and antifungal azoles are sold as cyclodextrin complexes. This includes nimesulide (Nimedex ®, Novartis), which is one of the model substances. Additional information is available in reference 3. Furthermore, the structures of these compounds are similar in a way since all of them contains two phenyl groups connected to each other directly or through an ether oxygen. To make the choice of model compounds clear to the reader, additional information has been added to the original text (lines 159-167).

The complexes obtained have a stoichiometry of 1:1, while in the text and in the figure caption there are other stoichiometries, they refer to mixtures and not complexes, although the authors call complexes mixtures in the ratio of 1:5.

The stoichiometric composition of the complexes shows the molar ratio of the host and guest in the complex. Each of the model compounds has a 1:1 ratio but the measurements were carried out at different molar concentration ratios of the host and guest compounds to shift the equilibrium toward the direction of complex formation (see our previous and here cited articles - References 21, 22). Nevertheless, to make this clear to the reader, we have corrected Figure captions 3 and 4 and Supplemntary Figure caption 1 and also in the Materials and Method section by writing “concentration ratio” instead of “ratio".

The introduction to the manuscript is too long and previous studies on the subject are not presented. Instead, there is a theoretical introduction as in the textbook.

As ICD spectroscopy is not commonly used to characterise cyclodextrin complexes, we aim to explain its benefits in a way that is clear to a wider readership. Previous studies are presented in several references, e.g. 8, 10, 20- 24 and 26-29.

The introduction does not explain the abbreviations (LADME, GMP and GLP). They are not in common use, like NMR, so they should be explained.

LADME has been changed to ‘pharmacokinetics’ (line 36), the GMP and GMP abbreviations are explained. (lines38-39)

The title of the manuscript ends with a period. We do not put periods at the end of titles.

It is corrected.

 The unit MHz, a unit of frequency, does not refer to a magnetic field, but to an electromagnetic wave (l. 58 and 59).

Indeed, the magnetic field strength is given in Tesla, we have corrected it (line 59)

 In l. 67, instead of "Cyd", the abbreviation "CyD" should be used.

It is corrected.

In the caption of Figure 3, the designation of the type of cyclodextrin is missing (l. 157).

It is corrected.

In l. 163, the comma after "Figure 4" should be omitted.

It is corrected.

In the figures in the main body of the manuscript and in the supporting materials, the stereochemistry on the C1-O bond should be indicated; since the geometry of the bonds is indicated at all, this should be done consistently.

We have changed the structure of fenbufen in the Supplementary Figure S1 so it is now consistent with the structure in Figure 2.

In the caption of Table 1, instead of the full name of the unit "debye" there should be the abbreviation "D".  

It is corrected.

In the supporting materials, correct the subscript in the water formula and in the deuterated methanol designation.

It is corrected.

Reviewer 2 Report

Comments and Suggestions for Authors

I really enjoyed reading this work and I think that eventually it should be published. The Authors have analyzed four CD inclusion complexes, using a method (Circular Dichroism) that is rarely used in this area. At least less frequently than other physicochemical methods. I really appreciate application of molecular modelling methods and NMR. However, I have also some questions and suggestions on how to improve this work.

Line 13, first, to „have a spectrum” is too colloquial and should be rewritten, second, this is not entirely true as you can record the spectrum of cyclodextrin but there will be no signals visible in it

Lines 20 and 132, in my opinion this is not a communication but a full-length article

Line 37, also as the taste masking agents as well

Line 49, here, the Authors describe the liquid state NMR, which should be clearly stated

This comment is not related to the current work but I would really enjoy reading the review, written by you, on the application of CD in the analysis of CyDs complexes. Please consider writing such a work in a future. I like your style and I can see that you are experts in this field.

Line 132, I really like that the aim is clearly stated. However, the information about the MM methods appear “out of nowhere”. While the Authors have described the analytical methods in the introduction, the haven’t even mentioned the computational methods. Therefore, I think, that a few sentences about the current status of QM studies and molecular dynamics simulations of cyclodextrin host–guest complexes with suitable references to recent reviews would be beneficial.

Line 138, please use plural

Figure 3, is it possible to use those results to determine the molar ratio between the host and guest?

Line 214, please remove “were”

Line 373, MD simulations. First, I think it would be beneficial to provide the structure of those water boxes, they can be put into the SI. Also, it is not clearly stated how many molecules of a drug and CyD per one water box were used. I assume only one for each case, is that correct?

Line 391-400, I think it’s great that the Authors have used the QM methods. However, I don’t know what exactly have you calculated? Energies (enthalpies) of complexation? NMR spectra? CD spectra? UV spectra? I can’t see any of those results presented in the current manuscript.

Lines 402-409, I think that this should be moved to the “discussion”. And, in the conclusions, the Authors should draw some general conclusions from their work.

Author Response

Thank you for your positive review. I would like to respond to your comments as follows:

I really enjoyed reading this work and I think that eventually it should be published. The Authors have analyzed four CD inclusion complexes, using a method (Circular Dichroism) that is rarely used in this area. At least less frequently than other physicochemical methods. I really appreciate application of molecular modelling methods and NMR. However, I have also some questions and suggestions on how to improve this work.

Line 13, first, to „have a spectrum” is too colloquial and should be rewritten, second, this is not entirely true as you can record the spectrum of cyclodextrin but there will be no signals visible in it.

The sentence has been rephrased as follows:

“For native and most of the substituted cyclodextrins this condition is not applicable because, although chiral, cyclodextrins lack a chromophore group and therefore have no characteristic CD spectra over 220 nm.”

Lines 20 and 132, in my opinion this is not a communication but a full-length article.

It is corrected.

Line 37, also as the taste masking agents as well

The corrected sentence:

“In medicinal products individual CyDs may be included as orphan drug [1] as processing excipients [2] or as elements affecting the pharmacokinetic properties of the active substance [3] and also as taste masking agents.”

Line 49, here, the Authors describe the liquid state NMR, which should be clearly stated.

It was corrected to: “Among the solution-phase analytical methods, liquid state NMR spectroscopy [10, 11], which can provide both qualitative and quantitative information, is a prominent example."

This comment is not related to the current work but I would really enjoy reading the review, written by you, on the application of CD in the analysis of CyDs complexes. Please consider writing such a work in a future. I like your style and I can see that you are experts in this field.

I appreciate your opinion and thank you for your encouragement.

Line 132, I really like that the aim is clearly stated. However, the information about the MM methods appear “out of nowhere”. While the Authors have described the analytical methods in the introduction, they haven’t even mentioned the computational methods. Therefore, I think, that a few sentences about the current status of QM studies and molecular dynamics simulations of cyclodextrin host–guest complexes with suitable references to recent reviews would be beneficial.

Thank you for your helpful comment, we have corrected the omission in lines 134-157 of the introduction. Two relevant recent studies have been cited. In our calculations the parameterisation was based on these articles.

  1. A. Mazurek et al. Application of Molecular Dynamics Simulations in the Analysis of Cyclodextrin Complexes. Int. J. Mol. Sci. 2021, 22, 9422.
  2. A. H. Mazurek & Łukasz Szeleszczuk. Current Status of Quantum Chemical Studies of Cyclodextrin Host–Guest Complexes. Molecules 2022, 27, 3874

Line 138, please use plural

It is corrected.

Figure 3, is it possible to use those results to determine the molar ratio between the host and guest?

If the ICD spectra obtained by varying the concentration of cyclodextrin show an isoelliptic point or the UV spectra recorded simultaneously show an isobestic point, it is certain that the complex is only 1:1. Further evidence can be provided by calculating the stability constant for intensity data read at a given wavelength from selected spectra using nonlinear parameter fitting. A high R2 (~1) value indicates a good fit. If the saturation curve displays anomalous behaviour, it suggests a different complex stoichiometry. Further information can be found in our previous article. [22]

Line 214, please remove “were”

It is now corrected.

Line 373, MD simulations. First, I think it would be beneficial to provide the structure of those water boxes, they can be put into the SI. Also, it is not clearly stated how many molecules of a drug and CyD per one water box were used. I assume only one for each case, is that correct?

The parameters for the water box are given in lines 408-409. The calculations were performed using only one host and one guest molecule, indeed (lines 429-431).

Line 391-400, I think it’s great that the Authors have used the QM methods. However, I don’t know what exactly have you calculated? Energies (enthalpies) of complexation? NMR spectra? CD spectra? UV spectra? I can’t see any of those results presented in the current manuscript.

The main objective was to calculate the transient dipole moment of the complexed molecules. This calculation can verify the sign of the ICD signal and the structural assumption based on it, but experimental results were necessary to validate the structural results obtained by the calculations. These results were provided by 2D ROESY NMR spectroscopy. Therefore, we have inserted a sentence at the beginning of section 2.3 (lines 239-240) to make the objective of the QM calculations clear to the reader: " Our objective is to calculate dipole vectors of the complex excited state and verify the sign of the ICD spectra based on these results."

Lines 402-409, I think that this should be moved to the “discussion”. And, in the conclusions, the Authors should draw some general conclusions from their work.

Following the suggestion, the previous conclusion has been relocated to the 3.3 subsection of the discussion chapter (lines 348-356), and a more general conclusion has been drawn (lines 455-467).

Reviewer 3 Report

Comments and Suggestions for Authors

In this manuscript, Kraszni et al. presented an interesting study about using Circular Dichroism (CD) spectroscopy to monitor the host-guest binding of a series of model compounds. The characterizations of the complex compound by 2D NMR unambiguously demonstrated the binding structures while the CD spectra gave quantitative information of the binding kinetics and thermodynamics. However, before recommending acceptance of this manuscript, there are a few issues to address properly. 

1. The rationale for choosing these model compounds, biphenyl and biphenyl ether should be more explicitly explained. 

2. It is difficult to correlate how the model structures can help us understand the Kodaka-Harata rules and how the theory is applied in this study. 

3. I can see how MD simulations can model the host-guest structure. But it should be feasible to obtain at least one single crystal structure, which is helpful in verifying the simulations. 

4. there are error signals in figure 11

5. Please double check if this work is not supported by any external funding. 

Author Response

In this manuscript, Kraszni et al. presented an interesting study about using Circular Dichroism (CD) spectroscopy to monitor the host-guest binding of a series of model compounds. The characterizations of the complex compound by 2D NMR unambiguously demonstrated the binding structures while the CD spectra gave quantitative information of the binding kinetics and thermodynamics. However, before recommending acceptance of this manuscript, there are a few issues to address properly. 

The rationale for choosing these model compounds, biphenyl and biphenyl ether should be more explicitly explained. 

The model compounds used are poorly water-soluble drug substances, which can have their solubility and bioavailability improved by cyclodextrin. Various non-steroidal anti-inflammatory drugs (NSAIDs) and antifungal azoles are sold as cyclodextrin complexes. This includes nimesulide (Nimedex ®, Novartis), which is one of the model substances. Furthermore, the structures of these compounds are similar in a way since all of them contains two phenyl groups connected to each other directly or through an ether oxygen. To make the choice of model compounds clear to the reader, additional information has been added to the original text (lines 159-167).

  1. It is difficult to correlate how the model structures can help us understand the Kodaka-Harata rules and how the theory is applied in this study. 

Figure 11 illustrates our answer to this question. The calculated structure is validated by 2D ROESY NMR data, which confirms that the direction of the calculated TDM vectors and the positive ICD signal align with the Kodaka-Harata rule (section 3.3, lines 348-356).

  1. I can see how MD simulations can model the host-guest structure. But it should be feasible to obtain at least one single crystal structure, which is helpful in verifying the simulations. 

Verification of the simulations were provided by 2D ROESY NMR spectroscopy. A solution phase structure is considered more dynamic than a single crystal. The NMR and ICD measurements provide signals for the most probable conformers with the highest probability of occurrence.

  1. there are error signals in figure 11

Please provide further clarification on the issue. We have not identified any errors in either the Word or PDF document.

  1. Please double check if this work is not supported by any external funding. 

The work was not supported by any grant, indeed.

Round 2

Reviewer 2 Report

Comments and Suggestions for Authors

The Authors have improved their work and current version can be published as it is now.